# A Study on the Psychological Noise Reduction Effect on Dental Handpiece Noise through the Bone Conduction Speaker Equipped Unit Chair and Notch-Filtered Music

Yeabon Jo [1,†], Woojin Kang [1,†], Sungwoo Hong [1], Joseph Vermont Bandoy [2], Hyuk-Sang Kwon [2], Heejung Kim [3,*] and Eunsung Song [1,*]

1   School of Integrated Technology, Gwangju Institute of Science and Technology, Gwangju 61005, Republic of Korea
2   Department of Biomedical Science and Engineering, Gwangju Institute of Science and Technology, Gwangju 61005, Republic of Korea
3   Chosun University Dental Hospital, Gwangju 61452, Republic of Korea
*   Correspondence: khjdds@chosun.ac.kr (H.K.); eunsung@gist.ac.kr (E.S.)
†   These authors contributed equally to this work.

**Abstract:** Anxiety in dental patients has caused inconvenient experiences during their dental visits due to the noise generated by the dental handpiece. High-frequency sounds generated by the handpiece have been challenging to reduce using the active control method that targets low-frequency sounds, as well as the difficulty in applying the noise control method using sound-absorbing materials, because the size of the handpiece is small. As an alternative, a method that can reduce noise and provide stability by playing music to patients is being studied. However, in most studies, there are inconveniences such as the need to turn the music volume higher to cover dental handpiece noise or having to wear headphones to play music. In this study, in order to reduce this inconvenience and optimize the noise reduction effect of music, we propose a technology that converts music into sound masking and a unit chair equipped with a bone conduction speaker that plays music, and through clinical trials with 35 patients, it was confirmed that the proposed system made the patients emotionally stable. In addition, by analyzing the causes of these emotional changes, it suggests that the preferred genre of music by patients should also be considered.

**Keywords:** sound masking; handpiece noise; unit chair; noise reduction; notch filter

## 1. Introduction

Clinical practice involving appropriate management in assisting patients' decisions can be challenging for patients with dental anxiety as it delays dental care utilization. Identifying anxiety triggers in dental patients can be useful in managing it effectively. Radiation coming from medical treatment is one of the most unpleasant experiences patients receive. A survey was conducted to know the emotional response of patients visiting a dental hospital about noise in a dental hospital. Results showed that most of them said that the noise bothered them (71.7%) and that they hesitated to visit the hospital (63.3%) because of it [1]. Noise coming from the dental handpiece was pointed out as the main cause of patients' anxiety. The sharp sound of the dental handpiece caused goosebumps (42.8%) and agitation (22.3%) [1]. High-speed handpieces are currently being improved and have stability against heat, pressure, and vibration while providing precision during treatment. These efforts would help manage anxiety in patients, most specifically when drilling the tooth [2].

Dental handpiece noise can be directly controlled using active and passive noise control methods. By using passive materials to absorb sound and vibration, passive noise management is a technique for reducing noise (sound-absorbing and insulating materials).

Applying the passive noise reduction method to dental handpiece noise management is challenging because passive materials need to be thicker and heavier [3]. Active noise control techniques are useful for reducing low-frequency noise [4], but they are challenging to implement for high-speed dental handpieces [5]. The direct noise reduction method therefore requires extensive studies taking into account the complicated characteristics of the handpiece noise. There is research that demonstrates the benefits of playing music to patients in reducing noise and anxiety [6–8]. Another study used music through speakers or a headset as a way to intervene in dental anxiety in patients [9]. However, various drawbacks were discovered in order for patients to receive indirect noise reduction effects, including the need for louder music than the sound of the dental handpiece, the need for the patients to wear headphones, and the disruption of patient and doctor communication [10]. In this study, we propose a sound masking technology specialized for dental handpiece noise to improve the psychological noise reduction effect through music. In the proposed system, a bone conduction speaker was mounted on the headrest so that the patient could listen to music immediately after lying on the unit chair, and sound masking technology was applied to the music to control the noise of the dental handpiece. The unit chair is a chair on which a patient can lie down to receive dental treatment and is equipped with various parts such as a dental handpiece and a dental operation light, but the possibility of using the headrest as a part of it has not been studied except for when the patient's head is rested [11]. Sound masking is a process in which an initial sound (called maskee) is overlaid with another sound (called masker) that can eliminate its component to make it less recognizable, affecting the auditory nerve stimulation of a person. Sound masking has been investigated and applied from insulating private office rooms [12] to masking industrial noise [13], but none yet for the noise generated by the dental handpiece. Sound masking technology can be divided into simultaneous masking (SM), which is a frequency-domain masking, and temporal masking (TM), which is a time-domain masking [14]. SM is a method in which the maskee sound and the masker sound are presented at the same time to shield the maskee sound, and TM is a method in which the other sound (maskee) in front or behind is shielded through a sudden sound stimulus (masker). Because music can be used as a masker due to the characteristics of continuous sound, the SM method was applied in this study.

The study aims to confirm the applicability of a headrest equipped with a bone conduction speaker and to confirm the psychological noise reduction effect of music converted for sound masking against dental handpiece noise. By applying a noise control system to a dental unit chair, we suggest the possibility of medical device-based service with on-site response capabilities for relieving patient fear and inducing psychological stability.

## 2. Methods

### 2.1. Development of Unit Chair with Bone Conduction Speaker

In the proposed system, as shown in Figure 1A, the unit chair's headrest has a bone conduction speaker attached to it so that the patient may hear the sound as soon as they sit in it for treatment. The bone conduction speakers (transducers) in the headrest are designed to reach both mastoids of the patient's head. In order for the bone conduction speaker to be able to reach the mastoid of the head, the component that holds the headrest head is split into two sides regardless of the size of the head (Figure 1C). Hearing can be affected by the bone conduction system depending on its distance from the user's head. The bone conduction speaker is placed on the mastoid part of the user's head and is divided into two sides of the headrest to cover the user's head using the weight of their head (Figure 1E). With this, patients can audibly hear the music coming from the bone conduction speaker [15]. By mounting the bone conduction speaker on the unit chair, patients can hear the sound without wearing headphones in the treatment room. Additionally, since the bone conduction speaker's sound cannot propagate in its surroundings, there is a possibility of using the device in isolation. This means that only the patient can hear the sound. A

bone conduction speaker system was selected since it has the benefit of allowing music to be listened to without causing a problem with the conversation with the dentist. As shown in Figure 1B,D, a bone conduction speaker system consists of a power supply, a main controller, an audio amplifier, a radio transceiver, an antenna, and a bone conduction transducer with a frequency range of 300 Hz to 19 kHz. It can output up to 2 W.

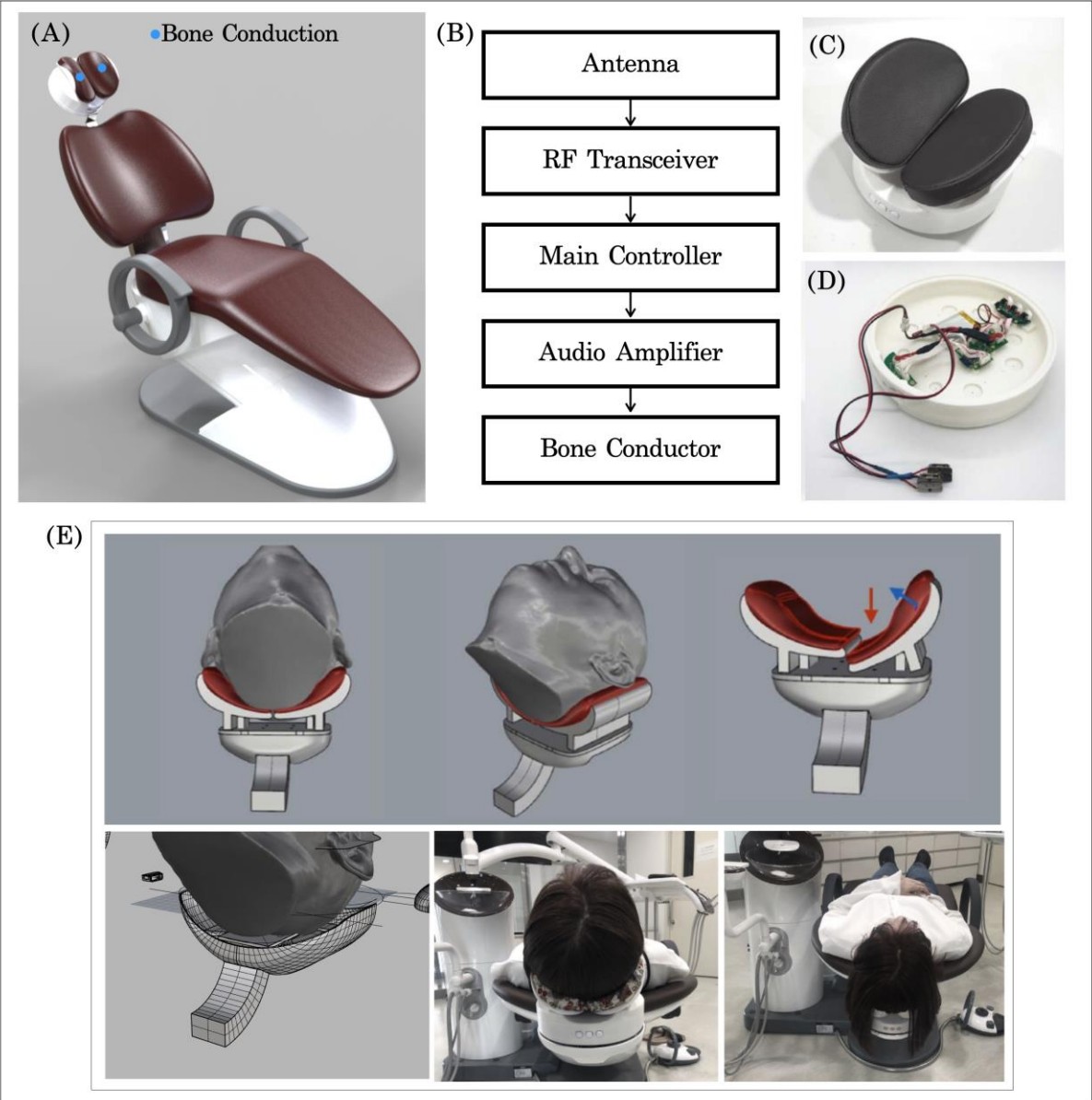

**Figure 1.** Configuration and photography of unit chair with bone conduction speaker: (**A**) unit chair with bone conduction; (**B**) diagram of bone conduction speaker; (**C**) headrest of unit chair; (**D**) bone conduction speaker system; (**E**) double-sided headrest designed to wrap around the head using the weight of the user's head.

## 2.2. Development of Notch-Filtered Music through Sound Masking Technology

Table 1 shows the equipment used for handpiece noise level measurement and analysis in this study.

**Table 1.** Equipment used for dental handpiece noise level measurement and analysis.

| Equipment | Company/Model |
| --- | --- |
| Dental Handpiece | TI-Max Z900 (NSK-Nakanishi International, Shimohinata, Japan) |
| Sound Level Meter | CK:161A (Cirrus, Sywell, UK) |
| Microphone | Zoom H6 and SSH-6 (Zoom, Tokyo, Japan) |
| Bone Conduction Transducer | BCE-1 22 × 14 mm$^2$ Bone Conducting Exciter (Dayton Audio, Springboro, OH, USA) |
| Amplifier | MH-M18 (SMG, Shanghai, China) |
| Audiometer | AMC493B (Larson Davids, Depew, NY, USA) |

To determine the masking threshold (dB(A)) to be used in clinical trials, the noise level generated in high-speed mode (360,000~430,000 rpm) was measured. Noise was measured in a closed treatment room with a background noise of less than 40 dB and a minimum background noise, and was measured at an ideal treatment distance of 30.0 cm (Figure 2A) [16].

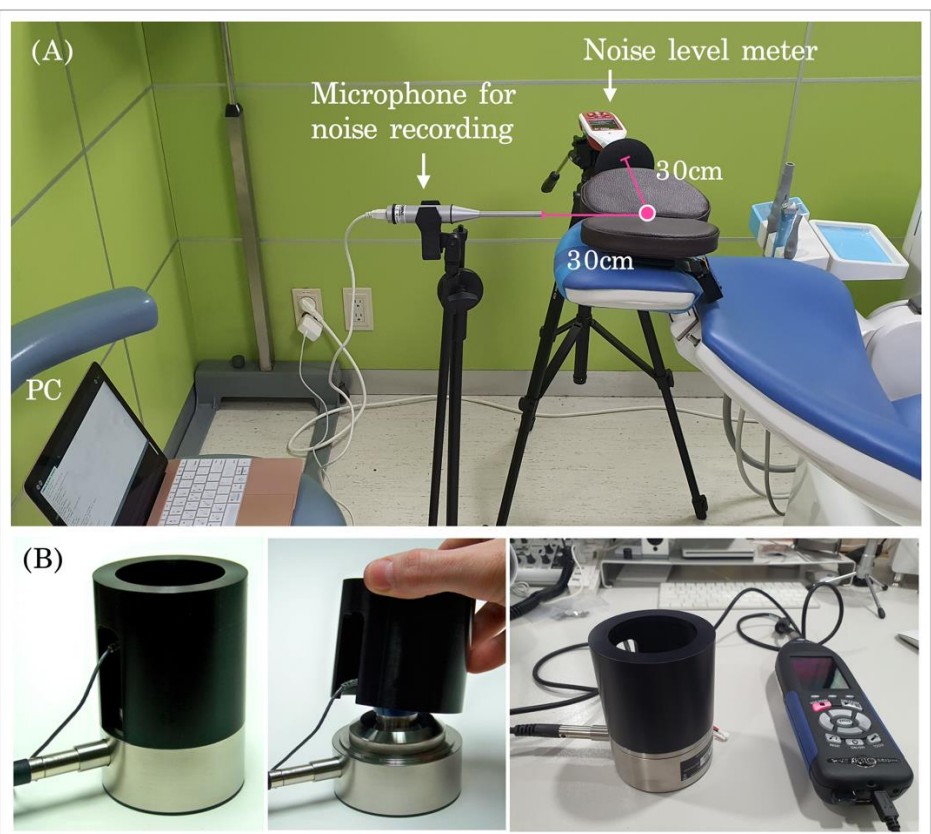

**Figure 2.** Apparatus and task method for the handpiece noise analyzing: (**A**) noise recording and measurement method; (**B**) calibration by audiometer.

The noise level was measured for 30 s, and the average sound pressure level exposed for 30 s was used as the measured value. Each experimental group was repeated 10 times under the same conditions to reduce measurement error. A measurement microphone was recorded for noise analysis, and the noise was analyzed by sampling the signal of the microphones with a frequency of up to 44.1 kHz. A spectral analysis was performed using the fast Fourier transform (FFT) algorithm to identify dominant frequencies. As shown in Figure 3A, the high-speed mode dental handpiece noise has peak frequencies of 6590 Hz and 13,180 Hz.

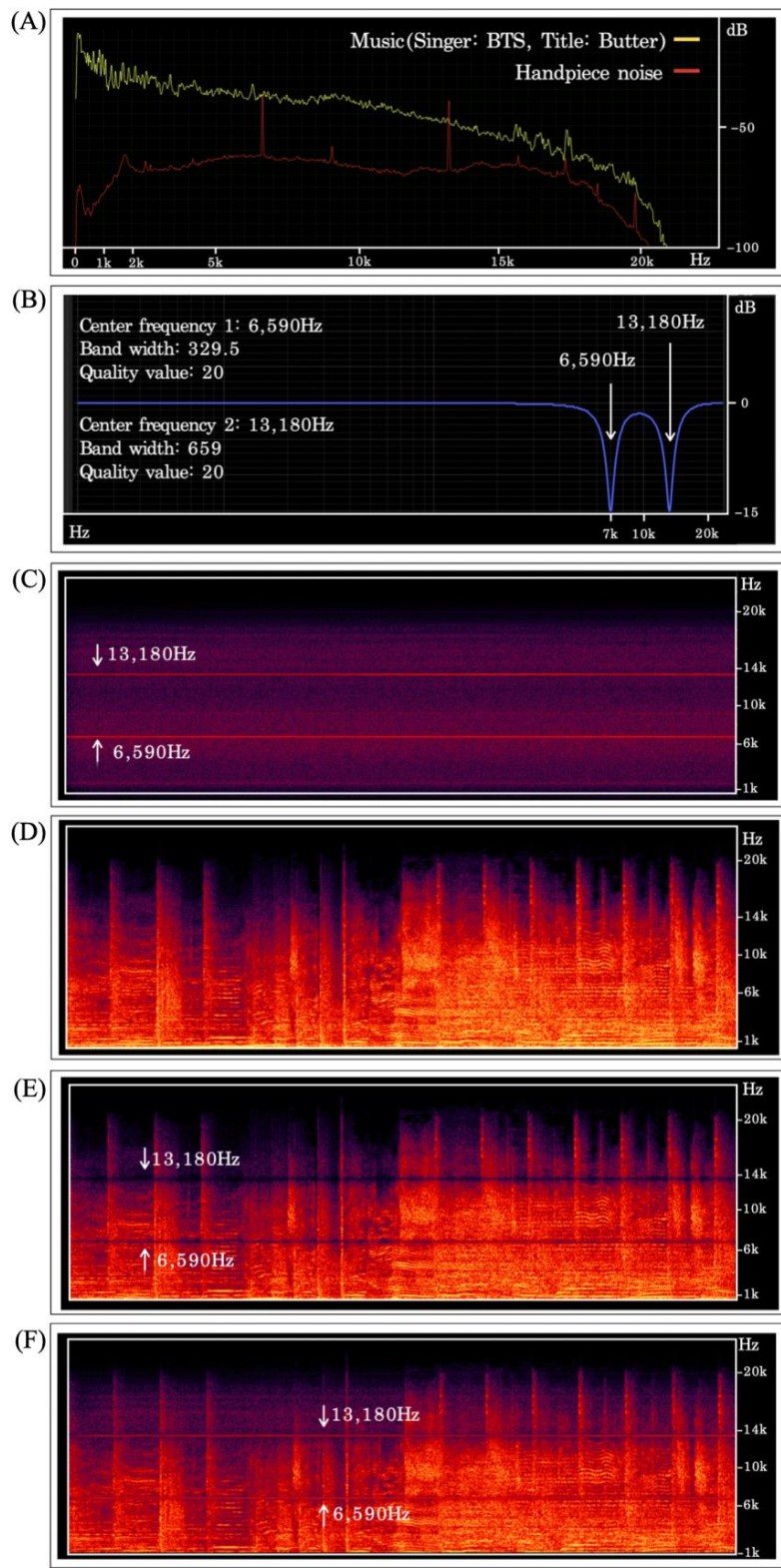

**Figure 3.** (**A**) Spectrogram analysis result of music and handpiece noise using FFT; (**B**) notch filter design; (**C**) high-speed mode dental handpiece noise; (**D**) music (singer: BTS, title: "Butter"); (**E**) music from which the target frequency of dental handpiece noise has been removed with notch filter; (**F**) synthesis of (**C,E**).

As a result of analyzing the characteristics of the bone conduction transducer used in the proposed system with a calibrated audiometer (Figure 2B), it was able to output

up to 96 dB(A). In clinical trials, since handpiece noise and music are played at a close distance to the patient, it is assumed that the loss is small even if each sound is transmitted differently through the airway and bone conduction. Therefore, the reference output of the bone conduction speaker was set equal to the masking threshold of the handpiece noise, which is 77 dB(A).

In this study, the notch-filtered music was used to develop the SM-type sound masking technology [17] (Figure 3B). To set the sound pressure of the music output through the bone conduction speaker and the center frequency of the notch filter, the masking threshold (dB) and target frequency (Hz) (Figure 3A) were used based on the dental handpiece noise analysis data (Figure 3C). In the frequency band of the patient's preferred music (Figure 3D), the target frequency was removed as shown in Figure 3E by using a notch filter.

### 2.3. Experiments

To verify the usability of the system proposed in this study to induce psychological noise reduction effects in patients, a subjective auditory evaluation was conducted on visiting patients at the Clinical Practice Training Center, College of Dentistry, Chosun University. For quantitative evaluation, 35 ordinary people in their 20s and 30s experienced the system in a dental clinic environment under the guidance of a dental specialist. Similar to the actual treatment, the noise was produced by driving the handpiece near the mouth while lying on the unit chair equipped with the handpiece noise masking system (Figure 4). Figure 5 shows the application method of masking technology and the noise control process to control handpiece noise transmitted into the air in this study. Music was played, and each subject heard the source randomly under the conditions shown in Table 2 in the unit chair. The music was played from each sample source for 3 min, and the patients rested for more than 5 min to reduce fatigue between listening to the sample sources. The patient responded to the perceived noise after the sample sounds source as preferred (the sound source intensity was set as a difference of $\pm 6$ dB based on the masking threshold of the handpiece noise (77 dB(A))). Compared to 77 dB, $\pm 6$ dB has a difference of $\pm 2$ times the sound pressure [18]. Normally, people are not good at detecting subtle differences in sound pressure, so we set it to $\pm 6$ dB so that people can clearly perceive the difference.

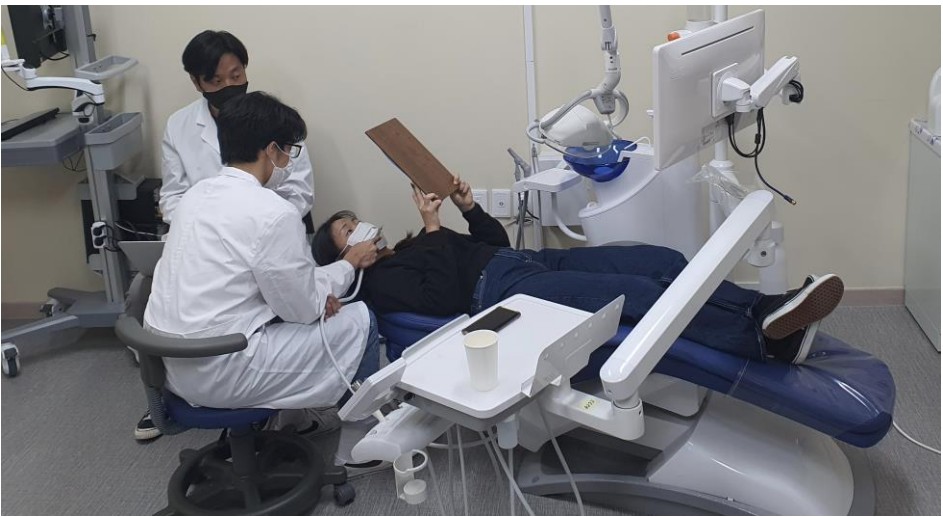

**Figure 4.** Clinical trial method of sound masking technology.

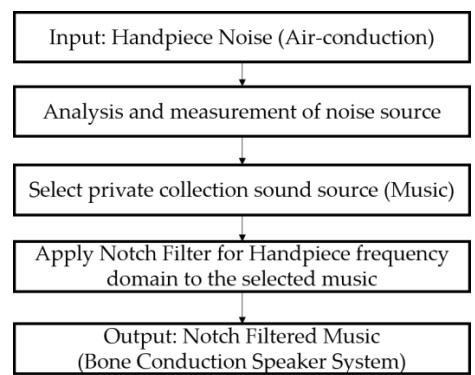

**Figure 5.** Noise masking process for experiments.

**Table 2.** The sound sources for clinical trial.

| Sound sources | A: (Base) Handpiece noise<br>B: Handpiece noise + Music<br>C: Handpiece noise + Notch-filtered music |
|---|---|
| Subjective noise intensity | −6 dB<br>±0 dB<br>+6 dB |

The patients took part in a survey that included questions based on their anxiety using a 7-point Likert scale after learning to listen to each sound source. In order to check the emotional changes of the patients for each sample sound source, the patients were asked to select the most similar vocabulary among the six basic emotional vocabularies (happiness, sadness, anger, fear, disgust, surprise) [19] that best describes their current emotions when listening to each sample sound source. After the patients responded to the survey, detailed reasons for their responses were collected.

## 3. Results

### 3.1. Survey Results of Noise Reduction Effect and Emotional Change

The effectiveness of the proposed system was confirmed through a survey. As shown in Figure 6, the patients preferred the volume of music (Sources B and C) to the dental handpiece noise level (base). Before the masking technology was applied, there were 20% of patients who preferred to listen to music at a low level (−6 dB), but after the masking technology was applied, only 3% preferred to listen to music at a low level. Overall, it was confirmed that people preferred to listen to louder music that is masked.

As shown in Figure 7, the patients felt an average anxiety level of 4.6 for Sound Source A, and an average anxiety level of 2.5 for Sound Sources B and C. As a result of statistical analysis using the Wilcoxon signed-rank t-test method, it was found that anxiety was reduced due to the psychological noise reduction effect when music was played to the patients (Sources B and C).

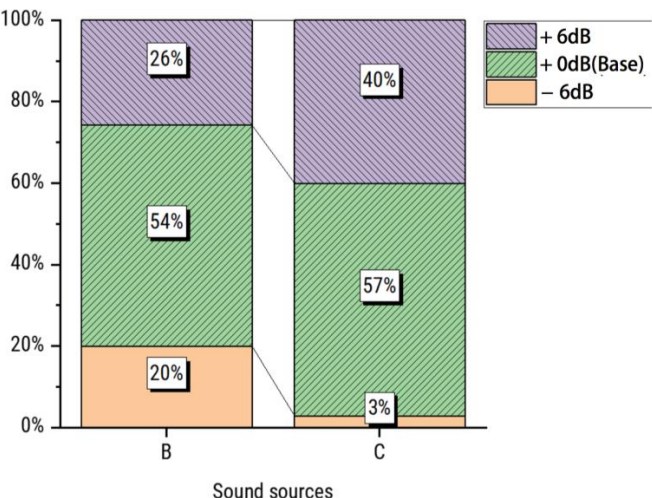

**Figure 6.** Subjective noise intensity results: (Source B) before applying masking technology (Source C); after applying masking technology.

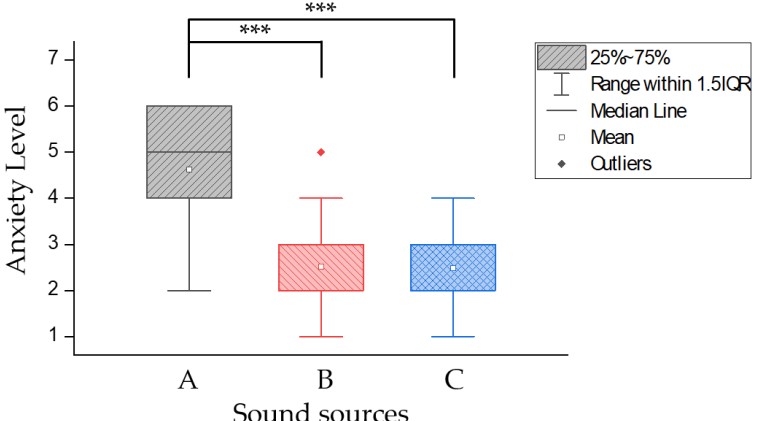

**Figure 7.** Results of patients' anxiety level evaluation for each sound source (7-point Likert scale) and statistical analysis results of Wilcoxon signed-rank *t*-test (*** $p < 0.001$).

### 3.2. Preference and Evoked Emotion Survey Results for Each Sound Source

As shown in Figure 8, when the dental handpiece noise (Source A) was played to the patients, negative emotions were recorded to have the most response (fear: 54.3%), but when the preferred music and dental handpiece noise were played together (Source B), the emotion of the respondents shifted to better feedback as they changed to positive emotions (Happiness 62.9%). When the patients were subjected to listening to the notch-filtered music (Source C), happiness was recorded to be 71.4%; compared to Sound Source B, happiness was improved by 8.5%, and negative emotions were reduced.

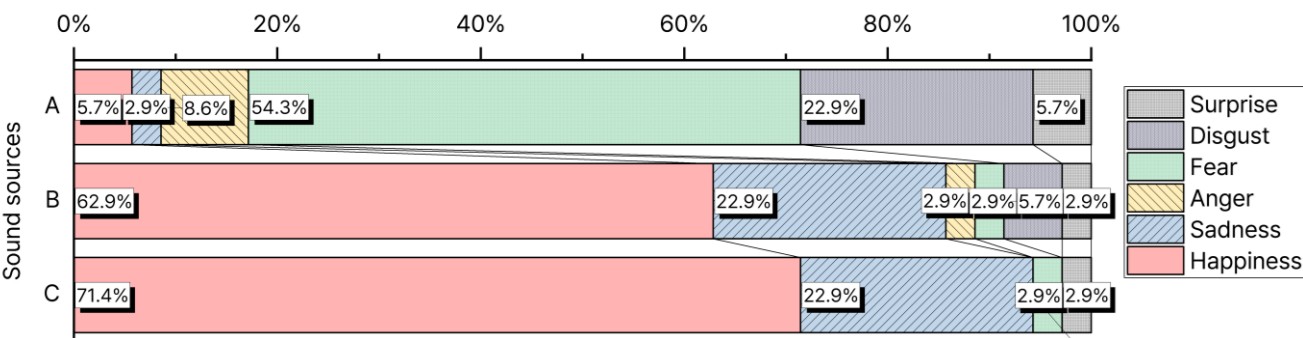

**Figure 8.** Results of the subjects' responses to the emotion questionnaire by sound sources.

## 4. Discussion

Contrary to expectations, a decrease in noise produced by the handpiece does not mean it lowers the anxiety level of patients, as louder music in other cases was preferred (Figure 6). The results of the interview showed that excessively loud music was not preferred, as it caused anxiety. However, patients preferred to listen to Sound Source C louder than Sound Source B because the notch filter used in sound masking technology removed some frequency bands of music and lowered the sound pressure by about 3.3 dB (Table 3: loudness). Total and average RMS amplitude shows the root mean square values of the selection. RMS values are based on the prevalence of specific amplitudes, often reflecting perceived loudness better than absolute or average amplitudes. Loudness shows the average amplitude, similar to RMS, but based on the LUFS scale, which is optimized for human perception (Table 3). It was confirmed through an interview with P1, such as "*I picked a louder sound to hear it better because there were parts I couldn't hear well compared to the original song*".

**Table 3.** Statistical data on average root mean square (RMS) and loudness change of the notch-filtered music.

|  | Music (Ex: BTS, "Butter") | Notch-Filtered Music |
| --- | --- | --- |
| **Total RMS Amplitude** | −15.89 dB | −18.36 dB |
| **Average RMS Amplitude** | −18.80 dB | −20.06 dB |
| **Loudness (Legacy)** | −14.36 dB | −17.61 dB |

It was found that notch-filtered music could induce positive emotions in the patients. In particular, the fact that the average anxiety values of Sound Sources B and C in Figure 7 are the same means that the proposed sound masking technique maintains the psychological noise reduction effect of music without reducing it. Compared to the noise of the handpiece (Sound Source A), it was observed that when the patients listened to music (Sound Sources B and C), the number of patients who felt the emotion of sadness increased. It was revealed through an interview that this was not a negative effect of sound masking technology, but rather an emotional characteristic of each music genre. In this study, the patients were given an option to freely select their preferred music, and some participants listened to the ballade genre song and evaluated it according to the music (Appendix A). It was confirmed that the proposed technique did not induce negative emotions, such as the interview with P4: "*I liked ballad music, so I felt sadness, but it was not an unpleasant feeling.*"

Like in previous studies [6–8], it was confirmed that although the patients could be induced to be emotionally stable when they listen to music, we found the volume or genre of music can have some influence on their emotions.

## 5. Conclusions

The study aims to control the noise generated by the dental handpiece by sound masking and a unit chair equipped with a bone conduction speaker. A clinical trial and

statistical analysis were conducted on 35 patients to investigate the possibility that the proposed system could help the emotional stability effects of music. In addition, analyzing the causes of the emotional changes implies that the patient's preferred genre of music may also be considered. However, this study conducted a noise control study focusing on air-conducted dental handpiece noise. In the future, we plan to conduct a noise control study on bone-conducted dental handpiece noise. In addition, in this paper, the user's subjective noise intensity was manually applied as a noise masking level through a questionnaire for clinical trials. Based on the experimental results, we plan to automate the application of the notch filters through real-time noise level and frequency analysis.

Through this study, it can be expected that this technique will be applied to improve the quality of medical services and to control noise (low frequency such as vibration) by other electronic devices as well as dental handpieces. In addition, it is thought that it will be helpful to improve patient satisfaction with treatment by finding and additionally applying factors that are expected to affect noise reduction measures during dental treatment in various ways.

**Author Contributions:** Conceptualization, E.S.; methodology, Y.J.; software, Y.J. and S.H.; validation, H.K.; formal analysis, Y.J.; investigation, W.K. and S.H.; resources, H.K.; data curation, Y.J.; writing—original draft preparation, Y.J.; writing—review and editing, J.V.B. and H.-S.K.; visualization, W.K.; supervision, E.S.; project administration, E.S.; funding acquisition, E.S. All authors have read and agreed to the published version of the manuscript.

**Funding:** This work was supported by the National Research Foundation of Korea (NRF) grant funded by the Korea government (MSIT) (No. 2021R1G1A1093763).

**Institutional Review Board Statement:** The study was conducted in accordance with the Declaration of Helsinki, and approved by the Institutional Review Board of Chosun University Dental Hospital, Korea (Protocol Code CUDHIRB 2104 003 and date of approval 17 August 2021).

**Informed Consent Statement:** Not applicable.

**Data Availability Statement:** Data are available upon request.

**Conflicts of Interest:** The authors declare no conflict of interest.

## Appendix A. Patient's Music Selection List

Appendix A shows the list of music selected by the patients in Table A1.

**Table A1.** Patient's music selection list.

| Patient No. | Title | Artist | Genre |
| --- | --- | --- | --- |
| P1 | My Universe | BTS and Coldplay | Rock |
| P2 | Modeun nal, modeun sungan | Paul Kim | Ballade |
| P3 | Next Level | Aespa | Dance |
| P4 | Butter | BTS | Dance |
| P5 | Back in Black | ACDC | Rock |
| P6 | Dambaegage agassi | YB | Rock |
| P7 | Eotteohge ibyeolkkaji Saranghagesseo | AKMU | Ballade |
| P8 | Sorry Sorry | Super Junior | Dance |
| P9 | Johannes Brahms, String Sextet No. 1 In B Flat Major, Op. 18 | Amadeus Quartet | Classic |
| P10 | Byeolboreo gaja | Juk Jae | Ballade |

**Table A1.** *Cont.*

| Patient No. | Title | Artist | Genre |
|---|---|---|---|
| P11 | Celebrity | IU | Dance |
| P12 | Garasadae | BewhY | Hip Hop |
| P13 | Georieseo | Sung Si Kyung | Ballade |
| P14 | Geujunge geudaereul manna | Lee Sun Hee | Ballade |
| P15 | Tornado of Souls | Megadeath | Rock |
| P16 | Singiru | C JAMM | Hip Hop |
| P17 | I Believe | Shin Seung Hun | Ballade |
| P18 | Tchaikovsky-String Quartet No. 1 In D Major, Op. 11 | Emerson String Quartet | Classic |
| P19 | Savage | Aespa | Dance |
| P20 | Stockholm Syndrome | Muse | Rock |
| P21 | Uptown Funk | Bruno Mars | Dance |
| P22 | I'm Not the Only One | Sam Smith | Ballade |
| P23 | No Song Without You | Honne | R&B |
| P24 | Nai | Yoon Jong-Shin | Ballade |
| P25 | Mommae | Jay Park | Hip Hop |
| P26 | Anajwo | Jung JoonIl | Ballade |
| P27 | Basket Case | Green Day | Rock |
| P28 | 200% | AKMU | Dance |
| P29 | Chobulhana | GOD | Ballade |
| P30 | Butter | BTS | Dance |
| P31 | Naege oneun gil | Sung Si Kyung | Ballade |
| P32 | Joheun nal | IU | Dance |
| P33 | Next Level | Aespa | Dance |
| P34 | Nakha | AKMU | Dance |
| P35 | Geokjeongmarayo Geudae | Juck Lee | Ballade |

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
