# Peer review of "A Study on the Psychological Noise Reduction Effect on Dental Handpiece Noise through the Bone Conduction Speaker Equipped Unit Chair and Notch-Filtered Music"

_applsci, doi:10.3390/app13010359_

Round 1

Reviewer 1 Report

The manuscript describes the study of the use of music with notch filtering function generated by bone conduction speaker equipped on the Dental chair to mask the noise radiated from the dental handpiece. This can reduce the psychosocially discomfort effect on the person. The content presents human perception based on questioning about the level of stress in 7-point Likert scale and six basic emotional vocabulary. 35 people are invited to attend the tests. The sound generated includes the original noise from dental handpiece, music sound without filter and with notch filtering of the peak frequencies generated by dental handpiece. The topic and findings are interesting and useful for engineers and scientists for further development on the product to reduce the human aural discomfort. Generally speaking, therefore, the current manuscript can be published if the authors make the amendments. There are several points which should be considered and revised for publication.

  1. Please mention the rhythm of the music you choose
  2. The authors have made effort on studying the effect of noise intensity of the music generated. What are the criteria you choose? Why do you choose ±6 dB?
  3. The authors use the level of stress by question the human. The level of stress actually depends on many factors. Can the author make justification on it? The term of using “level of stress” seems not appropriate.
  4. The authors did not describe the use of Table 3 but there are a lot of information.
  5. Figure 6 shows unclear information, please describe in manuscript in details.  

Reviewer 2 Report

The paper describes a method of noise reduction of a dental handpiece using a bone conduction loudspeaker. The study group consisted of 35 people. The findings of the authors are based on experimental measurements. Areas of strength: an interesting idea. Weaknesses: the results were not discussed with the literature data and previous research, stress level was presented unclearly, the handpiece noise was not produced by the real dental device.  I think that the experiment has been not designed correctly. The music which I do not like can be as irritating as the sound of handpiece.

Fig. 3:  The 6500Hz and 13000Hz annotations have probably been swapped, what in shown on horizontal axes?  Negative decibels - what was the reference value?   (B) - missing values on the vertical axis

133:  77 db (A)  - in Fig 3 is (-77) dB  - could you explain?

158: Why wasn't the handle sound produced by a real dental device? The handpiece induces bone conduction, as was stated in line 239.

166: Why 6 dB?

183 (Base) is not shown in Fig 6.

187: What was the method of stress measurement? What does it mean “Stress Level 5”?

192 According to Fig 7: “C” the method you have proposed is not effective.

208: How is this sentence related to Fig. 6?

229: Which studies? Why were they not used to compare with the results presented here?

235: I do not think so.

Has the phase difference between handpiece noise and notch filtered music been studied?

Round 2

Reviewer 2 Report

All my comments have been commented by authors.